

# Facilitating open-science with realistic fMRI simulation: validation and application

Cameron T. Ellis[1], Christopher Baldassano[2], Anna C. Schapiro[3], Ming Bo Cai[4] and Jonathan D. Cohen[4]

[1] Department of Psychology, Yale University, New Haven, CT, United States of America
[2] Department of Psychology, Columbia University, New York, NY, United States of America
[3] Department of Psychology, University of Pennsylvania, Philadelphia, PA, United States of America
[4] Princeton Neuroscience Institute, Princeton University, Princeton, NJ, United States of America

## ABSTRACT

With advances in methods for collecting and analyzing fMRI data, there is a concurrent need to understand how to reliably evaluate and optimally use these methods. Simulations of fMRI data can aid in both the evaluation of complex designs and the analysis of data. We present fmrisim, a new Python package for standardized, realistic simulation of fMRI data. This package is part of BrainIAK: a recently released open-source Python toolbox for advanced neuroimaging analyses. We describe how to use fmrisim to extract noise properties from real fMRI data and then create a synthetic dataset with matched noise properties and a user-specified signal. We validate the noise generated by fmrisim to show that it can approximate the noise properties of real data. We further show how fmrisim can help researchers find the optimal design in terms of power. The fmrisim package holds promise for improving the design of fMRI experiments, which may facilitate both the pre-registration of such experiments as well as the analysis of fMRI data.

Corresponding author
Cameron T. Ellis,
cameron.ellis@yale.edu

## INTRODUCTION

Over the past two decades, the use of functional Magnetic Resonance Imaging (fMRI) has exploded as a method for studying human brain function. In the early years of fMRI research, linear models were used to describe the voxel by voxel differences in activity between conditions (*Friston et al., 1995*). However, over time there has been a progressive shift toward methods that assess multivariate activity in order to probe representations and dynamics that are distributed throughout the brain (*McIntosh et al., 1996*; *Norman et al., 2006*). These new methods have taken advantage of high-performance computing and advanced software packages to answer questions that would otherwise be difficult to test with univariate procedures alone (*Cohen et al., 2017*). However, this shift has not been accompanied by commensurate advances in our understanding of how fMRI noise influences these analyses. Here we describe fmrisim, a new tool to address this need that
can help researchers evaluate and anticipate nuances that advanced neuroimaging analyses may introduce, and that can facilitate open science.

In the past, simulation has been a useful approach to addressing important issues regarding fMRI data. For example, simulations have been used to characterize the hemodynamic response function (HRF) in the analysis of event-related designs (*Burock et al., 1998*), as well as the spatial noise properties of fMRI data in cluster thresholding (*Forman et al., 1995*; *Ward, 2000*) and assessing inter-subject variability (*Erhardt et al., 2012*). However, others have noted (*Welvaert & Rosseel, 2014*) that the procedures for constructing these simulations are often not standard across studies and/or not clearly described. One remedy to this situation is neuRosim—an extensively developed toolbox in R that can be used to accurately simulate task-based activity, especially for univariate designs, in a standardized way across experiments (*Welvaert et al., 2011*). Another tool is STANCE, a MATLAB package that takes real data as a baseline and adds user-specified signal on top of it (*Hill et al., 2017*). simTB is another package that can match the spatial noise properties of its simulation to real data, and is particularly useful for group ICA simulations and connectivity analyses (*Erhardt et al., 2012*). These tools provide different approaches to simulating univariate brain activity, however they have been focused largely on univariate analyses. Similar tools are needed that address the effect of noise in multivariate analyses.

Synergistic with their use in data analysis, simulations can also facilitate open science and reproducibility (*Simonsohn, Nelson & Simmons, 2014*). Reproducibility is especially problematic in fMRI research, where researchers have innumerable preprocessing choices that can have important consequences for their results, but are not always carried out in a standard and/or clearly described manner. Although many remedies to this problem have been proposed, such as standardized preprocessing pipelines (*Esteban et al., 2019*), one potential solution is pre-registration (*Munafò et al., 2017*). Using services such as those offered by the Open Science Framework, researchers can specify in advance their design procedures and analysis pipeline and embargo it until review. Current guidelines for reproducible science encourage the pre-registration of both experimental design and analysis. This can be difficult for fMRI analyses: because of their scope and complexity it is sometimes hard to anticipate all of the possible concerns that otherwise may arise in analysis—especially those caused by noise—which often invites post-hoc analyses. However, the failure to provide algorithmically precise analysis plans in advance can inflate the risk of false discovery due to post-hoc exploration of different analysis pipelines.

We propose that simulations could facilitate pre-registration of fMRI data. Before any data is collected, researchers can simulate expected experiment effects embedded within a realistic model of fMRI noise, and construct a pipeline for analyzing the results that takes this into account. This would not only ensure explicit specification of the analysis procedure, but could provide more accurate and reliable estimates of effect sizes that can be used for power calculations. The result would be that interested readers could observe for themselves exactly what the researcher hypothesized and what analyses were planned in advance. This is not to discourage post-hoc, exploratory analyses; rather, it would simply

assist in demarcating what was predicted and what should be considered post-hoc analysis, so that the proper analyses and interpretations can be applied to each.

To be most effective, a simulator should be integrated with the tools used for experimental design and data analysis. While there are a number of packages used for standard fMRI analysis (e.g., FSL *Jenkinson et al., 2012*), and others for multivariate analysis (e.g., the Princeton MVPA toolbox *Detre et al., 2006*), there has recently been a large scale migration to Python for scientific computing, where tools such as Nilearn (*Abraham et al., 2014*) and Nipype (*Gorgolewski et al., 2011*) make it an increasingly attractive environment for fMRI data analysis. Among these is BrainIAK (Brain Imaging Analysis Kit, http://brainiak.org/), a newly released open-source Python toolbox that supports advanced neuroimaging analyses, such as multivariate pattern analysis (*Norman et al., 2006*), full-correlation matrix analysis (*Wang et al., 2015*), Inter-Subject Functional Connectivity (*Simony et al., 2016*), Bayesian Representational Similarity Analysis (*Cai et al., 2019*), Hierarchical Topographical Factor Analysis (*Manning et al., 2018*) and Shared Response Modeling (*Chen et al., 2015*). These methods are computationally intensive but have been optimized to exploit modern advances in high performance cluster computing. Thus, this represents a potentially valuable environment for a simulation package, and in particular one that addresses the impact of noise in these advanced, multivariate analysis methods. Ideally, this should be customizable, have appropriate defaults, and integrate seamlessly with existing tools for experiment design and analysis. Critically, it must be able to generate realistic, 'brain-like' data in order for it to work in a pre-registration pipeline.

In what follows we describe such a package—called fmrisim—that is a set of open-source Python functions integrated into the BrainIAK toolbox (*Kumar et al., 2019*). This package builds on the innovations of previous fMRI simulators, but focuses on its application to advanced, multivariate neuroimaging methods. The goal is to make realistic simulations of brain data that can be inserted into a standard fMRI analysis stream which makes use of multivariate methods. To achieve this, fmrisim employs a linear model that estimates and combines known sources of neural noise. Below we describe the steps necessary to generate a dataset appropriate for multivariate analysis. We then describe analyses to assess the quality of the simulation. Finally, we discuss applications of the simulator for evaluating the efficacy of experimental design and facilitating reproducibility.

## MATERIALS—fmrisim

fmrisim generates some of the known sources of fMRI noise and combines them to create simulations that share some of the properties of real fMRI data. Unlike other simulation tools, such as POSSUM (*Graham, Drobnjak & Zhang, 2016*), fmrisim is not a physics-based model but instead linearly combines a set of noise sources, inspired by biology and the physics of MRI, that are tuned in a data-driven fashion. We suggest that fmrisim is realistic in the sense that it approximates some, albeit not all, of the known properties of fMRI data (like simTB and STANCE), while also producing entirely novel volumetric data that can be preprocessed with the same tools that are used in typical fMRI analysis (like neuRosim).

fmrisim is developed under an Apache 2.0 license, meaning that it is free for distribution and modification for both commercial and non-commercial use. As part of the BrainIAK environment, updates and contributions to the package must pass an extensive suite of tests in order to validate that the changes do not compromise the quality of the tools.

The process of simulating participant data with fmrisim can be described in three steps: (1) specifying parameters, (2) generating noise, and (3) generating signal to add to the noise. Here we describe at a conceptual level what is needed to simulate a multivariate dataset with noise properties matched to a raw dataset. We encourage interested readers to go through the Jupyter Notebook in BrainIAK that demonstrates the relevant code to perform these steps (https://github.com/brainiak/brainiak/blob/master/examples/utils/fmrisim_multivariate_example.ipynb). The source code for fmrisim is extensively commented for accessibility.

## Specify parameters

In order to simulate fMRI data using fmrisim, the signal, noise, and acquisition parameters must be described. While all of these parameters can be defined by the user, some of these can instead be derived from real, pre-existing fMRI data and used to simulate data with similar attributes. These parameters, and which ones can be derived from real data, are described in the following section. In addition to these parameters, the simulation also needs a template that represents an approximate baseline of the fMRI signal to which noise can be added. This spatial volume is generated in fmrisim by averaging each voxel across time (*Welvaert et al., 2011*) and is used to create a binary mask of brain and non-brain voxels (*Abraham et al., 2014*). The template will also reflect any signal differences across the brain due to coil element strength. This template can be based on a standardized template, provided by default with fmrisim, or can be created from real fMRI data that was acquired by the user.

## Generate noise

fmrisim can generate various types of realistic fMRI noise. Much of the noise modeling was inspired by neuRosim (*Welvaert et al., 2011*). In fmrisim, a single function receives the specification of noise parameters and simulates whole brain data with noise properties approximating those parameters. Figure 1 shows examples of the noise types generated by fmrisim.

The noise of fMRI data is comprised of multiple components: drift; auto-regressive/moving-average; physiological; task-related noise and system noise. Drift and system noise are assumed to reflect machine-related noise and thus affect the entire field of view of the acquisition. The remaining components are assumed to be specific to the brain and thus have a smoothness component related to the smoothness of functional data. Using Gaussian random fields of a certain Full-Width Half-Max (FWHM), we can determine how smooth the temporal noise is across voxels (*Chumbley & Friston, 2009*). This volume of spatial noise is masked to only include brain voxels. Drift and system noise are added to all voxels in a volume.

To simulate drift, cosine basis functions of different phases are combined (Eq. (1)), with longer runs being comprised of more basis functions (*Welvaert et al., 2011*). The first basis
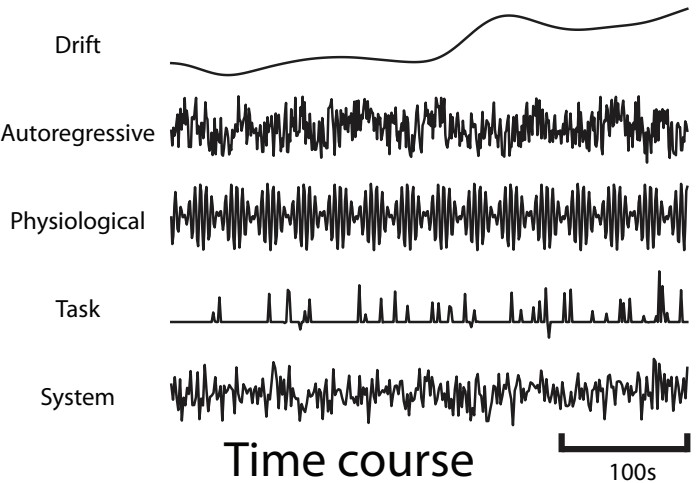

**Figure 1** **Example time courses of the different types of noise that are generated by fmrisim.** Each plot represents a voxel's activity for each type of noise.

function is a cosine basis function with a half cycle of the specified period. Each subsequent basis function has a frequency that is a multiple of the first basis function (akin to fmriprep *Esteban et al., 2019*), but their amplitudes diminish, such that 99% of the drift power is below the specified periodicity.

$$\sum_{i=1}^{\frac{L}{A}} \cos(i\pi \frac{t}{L} + p_i) r^{i-1} \tag{1}$$

where $L$ is the length in seconds of the run, $A$ is the volume acquisition time (AKA TR) in seconds, $i$ is the basis function counter, $t$ is the timestamp of each volume in seconds, $p_i$ is a random phase in radians for each iteration of $i$, $r$ is the proportion of drop off found by solving $0.99 = (1 - r^{\frac{2L}{P}})/(1 - r^{\frac{2L}{A}})$ for $r$, where $P$ is the periodicity (default is 150s).

Auto-regressive/moving-average (ARMA) noise is generated by creating a sequence of volumes, with each volume being generated by combining Gaussian noise with a specified proportion of the previous volumes (*Purdon & Weisskoff, 1998*). Precisely, the AR component specifies how much activity at each previous time point contributes to the activity at the current time point. The MA component does the same but for how much noise from previous time points contributes to the current time point (Eq. (2)). Note that AR is used to model noise because it will reflect the BOLD activity in regions that do not contain task-related signal. fmrisim provides tools to insert task-related signal with appropriate AR dynamics (section 'Generate signal and add it to noise').

$$Y_t = X_t + \varphi Y_{t-1} + \omega X_{t-1} \tag{2}$$

where $Y$ is the simulated noise at time point $t$, $X$ is the component of new 3-dimensional Gaussian noise, $\varphi$ is the auto-regressive weight, and $\omega$ is the moving average weight.

Physiological noise is modeled by combining sine waves comprised of heart rate (1.17 Hz) and respiration rate (0.2 Hz) (*Biswal, Deyoe & Hyde, 1996*) with random phase. Finally, task-related noise is simulated by adding Gaussian or Rician noise to time points where there are events (determined by the design of the experiment). The ARMA, physiological and task-related noise components are mixed together according to user-defined parameters and then set to the appropriate magnitude of temporal variance (Eq. (3)), derived from the Signal-to-Fluctuation-Noise Ratio (SFNR). The SFNR, also known as temporal signal-to-noise, reflects how much temporal variation there is in brain voxels relative to their mean activity (*Friedman & Glover, 2006*). If the SFNR is high then the amount of temporal variation will be low and thus so will these temporal noise components. Drift is weighted separately and added after this.

$$W\left(w_{ARMA}v_{ARMA} + w_{physio}v_{physio} + w_{task}v_{task}\right) + w_{drift}v_{drift} \tag{3}$$

where $W$ is the brain-specific noise that is determined by the standard deviation of the detrended temporal variability, averaged across the brain, $w$ is a weight from 0 to 1 set by the user or fit on the data ($w$ for ARMA, physiology and task-based noise must sum to 1), and $v$ is a 4-dimensional volume of normalized noise.

The volume created by the combination of these noise components is then added with system noise. System noise results from heat-related motion in MRI scans (*Bodurka et al., 2007*). System noise is Rician (*Gudbjartsson & Patz, 1995*); however, we have determined that the distribution of voxel intensity in non-brain regions (including the skull and eyes which are sensitive to T2* measurements) is also approximately a Rician distribution (e.g., the eyes have high MR values but the vast majority of voxels are near zero). As stated above, fmrisim creates a template of voxel activity (both brain and non-brain voxels) to which the simulated noise is added. This template also has non-brain voxels with a Rician distribution, so adding Rician noise to voxels with an already Rician distribution leads to inappropriately 'spiky' data. Instead, we have found that system noise is better approximated by Gaussian noise added to the template of average voxel activity. The magnitude of system noise is determined by both a spatial noise component, which depends on the Signal to Noise Ratio (SNR) value, and a temporal component of system noise, that depends on the SFNR.

All the noise parameters, like SNR, can be manually specified; however, it is also possible to extract some of these noise parameters directly from an fMRI dataset using a single automated function in fmrisim. Specifically, fmrisim utilizes tools for estimating the SNR, SFNR, FWHM and auto-regressive noise properties of real data. This real data can come from a sample of previously collected fMRI data that you believe will have similar attributes (e.g., acquisition, signal to noise quality, etc.) to the data you wish to simulate. To calculate the SNR, fmrisim compares brain signal with non-brain spatial variance. To do this, the activity in brain voxels is spatially averaged for the middle time point and divided by the standard deviation in activity across non-brain voxels for that same middle time point (*Triantafyllou et al., 2005*). To estimate SFNR, fmrisim divides each voxel's mean activity by the standard deviation of its activity over time, after it has been detrended with a second order polynomial. This is done for every brain voxel and then averaged to give an estimate of the global SFNR. The FWHM of a volume is calculated by first computing variance in the

X, Y, and Z dimensions, converting this to FWHM for each dimension and then averaging both across each dimension and across a sample of time points. Finally, fmrisim estimates ARMA using the statsmodels Python package by assuming an AR order of 1 and an MA order of 1 (i.e., consider only the previous time point). This is calculated independently for 100 voxels and then averaged to give the estimate for a participant.

The noise estimates of real fMRI data provided by the fmrisim tools described above can then be used to simulate data with equivalent noise parameters; however, there are some noise properties that fmrisim does not estimate. This is because they are otherwise trivially dealt with in preprocessing (e.g., drift), or they are impossible to retrieve from the raw data without knowing the latent signal that generated the brain data (e.g., task noise) or without greater temporal precision (e.g., physiological noise). These types of noise must be specified manually in order for them to be used by fmrisim.

Importantly, estimating noise parameters from empirical data depends on assumptions about the appropriate noise properties of fMRI data and how they interact. If the data is already preprocessed then this can interfere with the way noise properties ought to be inferred. For instance, if all non-brain voxels have been masked, then these zeroed-out voxels will be an inappropriate baseline for non-brain variability. In addition, because of the stochasticity of the simulation, the estimation is not fully invertible: simulating a brain with a set of noise parameters and then estimating the parameters from that simulated brain will recover ones that are similar, but not necessarily identical, to those originally specified. To address this, and generate simulations with noise characteristics as close to the ones specified as possible, fmrisim iteratively simulates and checks the noise parameters until they fall within a specified tolerance of what was specified. This is useful for matching noise parameters to a specific participant, not just to typical human fMRI data. The fitting tolerance is by default set to within 5% (i.e., the difference between each real and simulated noise parameter is less than 5%). This can be made stricter by the user if desired; however there is inherent stochasticity in the noise generation process, especially for ARMA and system noise, that makes it less likely to converge when the tolerance is set lower. Once the noise parameters are estimated, they are then used to generate time series data to which one or more hypothesized neural signals are added.

## Generate signal and add it to noise

fmrisim has tools to simulate many different types of neural signal. These tools are useful for generating signals intended to represent patterns of activation in specific regions of interest (ROIs). However, custom scripts may be necessary for more complex signals, such as activity coupling between regions for connectivity analyses. To design signal activity, it is necessary to establish the predicted effect size, the ROI/spatial extent of the signal, and its time course. For instance, the notebook describes a case in which two conditions evoke different patterns of activity in the same set of voxels in the brain. This pattern does not manifest as a uniform change in voxel activity across the voxels containing signal (i.e., the mean evoked activity is not different between conditions). Instead, each trial of a condition evokes an independent pattern across voxels. This activity can then be convolved with the hemodynamic response in order to estimate the predicted pattern of measured activity.

fmrisim also makes it easy to calibrate the magnitude of a signal using various metrics, such as percent signal change or contrast to noise ratio (*Poldrack, Mumford & Nichols, 2011*; *Welvaert & Rosseel, 2013*).

The final step is to combine, through addition, the noise volume with the signal volume to create a brain that is ready for preprocessing and analysis.

### Expected background for fmrisim users

fmrisim was designed with the goal of being intuitive and user-friendly in order to support the adoption of simulation in the design and analysis of fMRI experiments. We expect that users who are Python novices will understand the provided notebooks that show how the simulator works. In order to use the simulator, no specialized mathematical or statistical background is required; although familiarity with fMRI analysis is expected. For users to design and simulate their own experiments, they need only basic understanding of Python scripting in order to adapt the provided examples. Such use of fmrisim has been validated as part of a university course (*Kumar et al., 2019*), in which intermediate Python users were able to adapt the simulation code provided in various ways. If users run into issues with fmrisim, they are encouraged to reach out through the BrainIAK forum.

## DISCUSSION

Above we described how fmrisim generates realistic fMRI data. For more detail we point readers to the documentation, and to the online notebook that demonstrates the implementation of a simulation to see how the tool is actually used. fmrisim is intended to be flexible, allowing for the generation and specification of many different types of fMRI noise and signal. Of course, the simulator is not perfect: a complete simulation would require an understanding of all of the latent causal interactions in the brain and how these manifest in fMRI data—a goal that remains the focus of the research enterprise itself.

fmrisim is limited in the scope of noise that it attempts to simulate. There are likely many sources of noise that contribute to fMRI noise, but remain poorly understood or undetected. Moreover, there are known types of noise that fmrisim does not attempt to model (e.g., higher order structure inherent in baseline or resting state activity; *Raichle et al., 2001*) and fmrisim applies noise in a linear manner despite what we know about the non-linear dynamics of the brain. Adding additional noise structure in future versions of this package could make it possible to simulate functional connectivity, as other simulation packages have done successfully (*Allen et al., 2014*).

Motion has substantial effects on fMRI data quality but is not accounted for here. Other simulations have modeled the effects of rigid body transformations (*Allen et al., 2014*) and motion between slice acquisitions (*Jones, Bandettini & Birn, 2008*), but these effects of motion can be removed with linear motion correction software. The more pernicious effects of motion are harder to simulate, such as the non-linear changes to T1 relaxation across planes (*Friston et al., 1996*), and are not addressed in this package. Other global effects on fMRI signal, like sharp inhalation or breath holding (*Power et al., 2017*), are also not modeled here but could be in future releases.

fmrisim is also limited in that assumes the distribution of noise is uniform throughout the brain, as determined by a 3-dimensional Gaussian. This assumption is incorrect since different types of noise have stronger effects in different parts of the brain. For instance, physiological noise has greater effects near major arteries and ventricles. It might be possible to introduce non-uniform noise if we assumed a standard template (and fmrisim supports this currently) but to do this at a participant-specific level would require extensive preprocessing to locate the regions of a participant's brain that are sensitive to certain types of noise. Moreover, the slice acquisition protocol, be it uni-band or multi-band, will introduce susceptibility to noise that may vary by slice. This is not implemented explicitly in fmrisim; nonetheless, slice specific differences are implicitly included in the brain templates.

These limitations of fmrisim prevent it from being a fully realistic simulation of fMRI data; however, the use of fmrisim to simulate at least some of the known properties of fMRI data holds promise for improving experimental design and data analysis, as illustrated in the examples below.

## Experiment 1—validation of fmrisim

To evaluate the validity of the simulator, we test whether various noise parameters of the simulated data match the noise parameters of real data. All the code for these analyses and plots is available online (https://github.com/CameronTEllis/fmrisim_validation_application.git).

### *Materials and methods*

For these analyses we used the publicly available Corr_MVPA dataset (*Bejjanki et al., 2017*: http://arks.princeton.edu/ark:/88435/dsp01dn39x4181). This dataset contains two task runs, two rest runs and an anatomical image for each of 17 participants. Participants provided written consent in accordance with the institutional review board of Princeton University. This is appropriate for our analyses because it is a typical sized dataset with no preprocessing. For this data set we only considered the rest runs. These were collected on a 3T scanner (Siemens Skyra) with a 16 channel head coil and a T2* gradient-echo planar imaging sequence (TR = 1.5 s, TE = 28 ms, flip angle = 64°, matrix = 64 × 64, slices = 27, resolution = 3 × 3 × 3.5 mm).

We compared the noise properties of the real and simulated data to test the efficacy of fmrisim. To do this, the noise properties of the participant data were extracted and then used to create simulated data. We then estimated the noise properties of the simulated data, including SNR, SFNR, auto-regression, and FWHM. The noise properties of the original data were then compared with those measured in the simulated data in order to determine the fidelity of the simulation. In other words, we tested whether the known noise properties utilized by the simulator were similar to the noise properties estimated from the data generated by the simulator. We used the default fitting procedure and created 10 simulations for the rest runs of each participant in order to measure the reliability of each simulation.

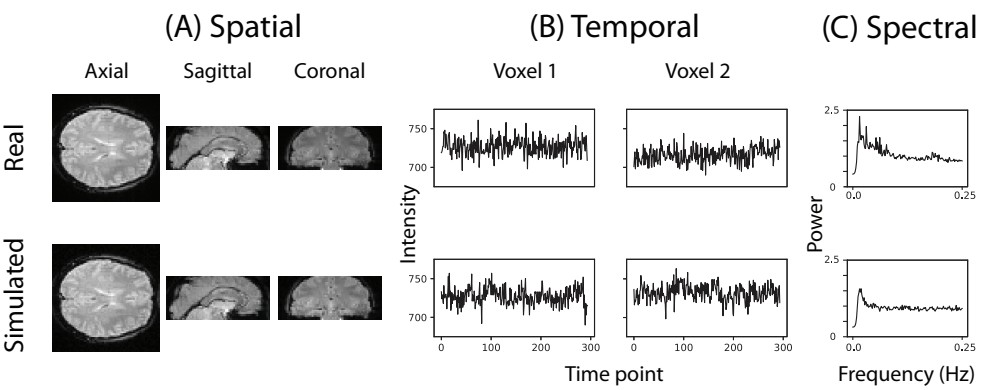

**Figure 2 Example of the spatial and temporal structure of real and simulated data.** (A) depicts the spatial structure of real data (top) and fitted simulated data (bottom). (B) shows the time course of sample voxels and (C) shows the power spectra of a sample of high-pass filtered voxels.

## Results

Figure 2 illustrates that the simulator can at least superficially approximate both the spatial and temporal properties of real fMRI data. Figure 2C shows the averaged power spectra for 1,000 randomly sampled voxels after they have been high-pass filtered (100s cut-off, butter filter). This suggests that the real and simulated voxels have similar spectral properties.

To evaluate whether fmrisim can accurately simulate brain data with specific noise parameters, we compared the noise parameters estimated from the real and simulated data. The ideal pattern of results is that there is no difference between the real and simulated data. Figure 3 shows boxplots of the noise parameters of the real and simulated data across participants. Notably, the majority of simulated brains (across all participants, runs and resamples) are within 5% of the target parameter (proportion within 5% for each noise component: SNR = 98.5%, SFNR = 100%, Auto-Regression = 92.1%, FWHM = 100%). Although advantageous, fitting is not perfect because of the sequential nature of the fitting process (e.g., SNR is fit before autoregressive noise) and the inherent randomness in the simulation.

Not depicted in Fig. 3 is the variance in the estimated noise parameters on each simulation. Although each simulation has inherent randomness, the variance in estimates is small relative to the variance between individuals with different noise parameters (mean variance of noise parameters extracted from simulations: SNR: $M_{Var} = 0.008$, SFNR: $M_{Var} = 1.428$, Auto-Regression: $M_{Var} = 0.0006$, FWHM: $M_{Var} = 0.0002$). Execution times for this data set were reasonable: the simulation with fitting took 278.3 s (SD = 39.8 s, max = 381.9 s) on average to complete for each participant/run on a single core of an Intel Xeon E5 processor (8 cores, 2.6 GHz).

## Discussion

We evaluated how well fmrisim is able to recreate the noise properties of fMRI data and showed that it can do so adequately. Critically, the noise parameter estimates were participant/run specific, which means that fmrisim can simulate data with noise properties

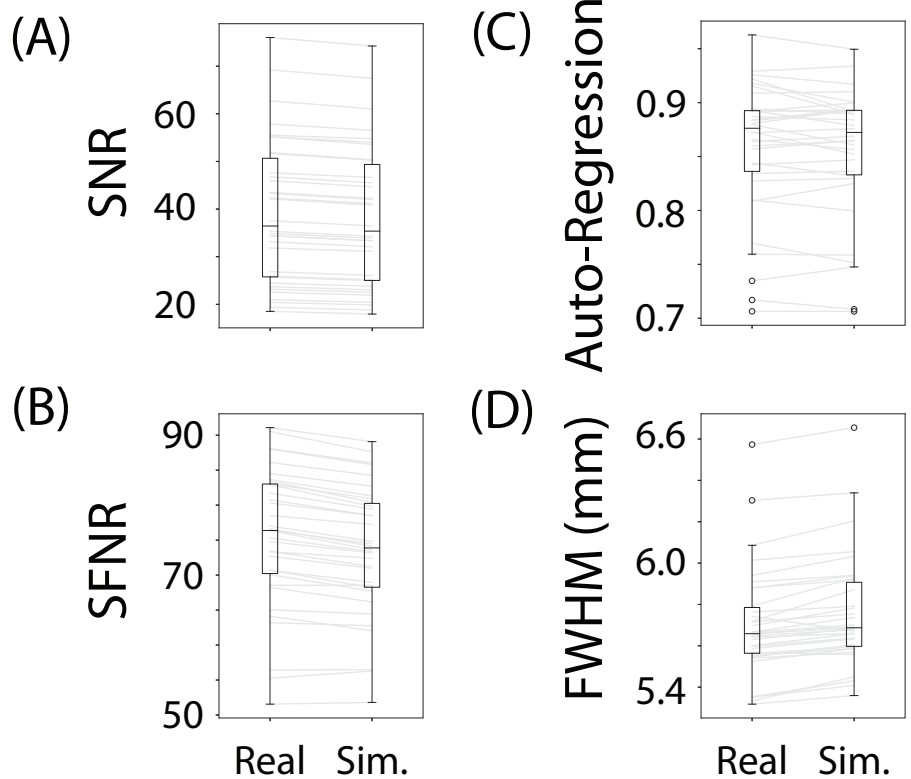

**Figure 3 Boxplots of noise parameters from real data and fitted simulated data.** For each participant and run, 10 simulations of their real data were generated and their noise parameters estimated. These noise parameters were then averaged within participant/run and plotted, and compared with the noise parameters from their corresponding real data. The grey lines connect the participant/run parameter value, the ellipses refer to outliers. The noise parameters tested were (A) Signal-to-Noise Ratio, (B) Signal-to-Fluctuation-Noise Ratio, (C) Auto-Regression, (D) Full-Width Half-Max.

matched to individual participants. In this sense, we claim that fmrisim is realistic; however, there are likely numerous nuanced and non-linear noise properties of real data not captured by fmrisim. Moreover, the simulations described above are necessarily incomplete, since known and important sources of noise, such as physiological noise (*Raj, Anderson & Gore, 2001*), could not be estimated from the data. Nonetheless, we believe fmrisim can simulate many of the critical properties that are necessary for optimizing experiment design.

Importantly, the tests outlined above reflect a lower bound of what an fMRI simulator is able to do. Future improvements to fmrisim or other simulators should capture more than just the descriptive statistics of noise in fMRI data but should also reflect the latent noise generation properties of the brain. Moreover, future improvements should include realistic simulations of motion artifacts and global signal changes. Nevertheless, fmrisim provides a useful advance in the ability to evaluate how fMRI noise and signal interact, that can be used for experimental design and analysis. We demonstrate such potential in the example application described below.

## Experiment 2—application of fmrisim

The interactions between experiment design, noise characteristics of the data, and their consequences on statistical power can be complex and difficult to anticipate without appropriate tools. Here, we describe how fmrisim can be used to optimally design an experiment for maximizing statistical power.

Statistical power, in the context of null hypothesis testing, reflects the likelihood of getting a significant result from a sample, assuming that there is a real effect to be found. Designing experiments with sufficient power is critical for conducting reliable research (*Cohen, 1992*). Power tests are important at the design stage for estimating how the number of participants or trials impacts the likelihood of reaching significance (*Poldrack, Mumford & Nichols, 2011*). However, they can also be useful after data collection for determining the likelihood of a given result (*Button et al., 2013*).

Power tests are particularly important for fMRI research: data acquisition is expensive, sample sizes are small, and effect sizes can be weak. Statistical power in fMRI research is affected by the size and character of the signal to be measured (which can vary widely across brain region; *Gläscher, 2009*; *Desmond & Glover, 2002*), the number of participants that will be studied, and the threshold for significance. Typically, power has been assessed in fMRI studies with reference to previous research with similar designs, using these to predict the effect size for that design. However, in cases for which such information is not available (e.g., in novel designs and/or populations), simulations provide a valuable alternative approach: by simulating a hypothesized signal in the brain, it is possible to estimate how many trials, participants, etc., are needed to pass a significance threshold for the analysis of interest.

One aspect of fMRI design for which power issues have been explored is in selecting the inter-stimulus interval (ISI). To maximize power in task-based fMRI, it is considered best to have slow designs in which events are spaced out to mitigate the overlap of the hemodynamic response (*Friston et al., 1999*). However, sometimes the only way to evaluate a research question is to use faster event-related designs, in which events occur within 10s of each other, either because a slower design would be impractically long or because the cognitive process requires faster paced stimuli and/or responses. Although fast designs mean the brain's slow hemodynamic response overlaps more across events, a shorter ISI can allow for more trials (i.e., the efficiency is increased), which can increase the statistical power for both univariate (*Dale, 1999*) and multivariate (*Kriegeskorte, Mur & Bandettini, 2008*) analyses. This is especially true when the time between events is randomly jittered (*Burock et al., 1998*), as different combinations of overlap can be sampled to improve deconvolution. Moreover, if conditions are presented multiple times then it is considered a good design choice to present them in a randomized order, since it will result in different overlaps between evoked representations (*Burock et al., 1998*). Thus, ISI can be a critical factor in determining the effect size and power of the design.

Although longer ISIs with randomized event sequences are generally considered to optimize power, most demonstrations of this have been limited to univariate analyses (although see *Kriegeskorte, Mur & Bandettini (2008)*). However, several factors could complicate, and potentially invalidate generalization of these findings to multivariate

analyses. For instance, calculating efficiency and covariance are important and tractable within univariate analyses, in which different voxels are treated as representing different stimulus conditions; however, doing so becomes much more complex if an effect is multivariate—that is, a single voxel is involved in some or all of the stimulus conditions.

To evaluate the effects of ISI and some related design choices on power, we used fmrisim with data from a study by *Schapiro et al. (2013)*. Below, we first describe use of fmrisim to observe what signal magnitude is necessary to match their result. We then show how different design choices can influence the expected effect size. This provides an example not only of how fmrisim can be used to explore the power of experimental designs, but also how this facilitates open science by making it easy for researchers to pre-register experiment analyses and clearly demarcate planned versus post-hoc analyses. All of the analyses described in what follows, as well as the necessary data to perform them, are available online (https://github.com/CameronTEllis/fmrisim_validation_application.git).

### Materials

*Schapiro et al. (2013)* examined how event segmentation is achieved in the brain. Figure 4 shows a graph representing the 15 stimuli participants saw in this study and the possible transitions (grey) between stimuli during the learning phase of the experiment. In their fMRI experiment, the stimuli were fractal-like images presented for 1s with a 1, 3, or 5s ISI. Although the assignment of images to graph nodes was random, the structured temporal transitions between these stimuli were expected to change the image representations to reflect the graph community structure (*Fortunato, 2010*). As participants observed a random walk through the possible transitions, the graph structure tends to produce long sequences of stimuli from within one community before transitioning to another community. Transitions between communities were only possible by passing through certain nodes on the graph. Behavioral evidence suggested that over the course of exposure, participants come to represent the transitions between communities as event boundaries.

Participants were scanned while observing the sequences of stimuli, and the imaging data were used to examine the similarity structure in the neural representations of the stimuli. Participants provided written consent in accordance with the institutional review board of Princeton University. fMRI data were collected in a 3T scanner (Siemens Allegra) with a 16 channel head coil and a T2* gradient-echo planar imaging sequence (TR = 2s, TE = 30 ms, flip angle = 90°, matrix = 64 × 64, slices = 34, resolution = 3 × 3 × 3 mm, gap = 1 mm). After participants had 35 min of exposure to random walks on the graph outside the scanner, they viewed alternations between random walks and Hamiltonian paths (i.e., every node visited exactly once) in the scanner. One particular Hamiltonian path was chosen for each subject (e.g., the perimeter around the dots in Fig. 4) and was presented in clockwise and counterclockwise directions. In order to avoid potentially confounding item repetition effects in the random walks, the first three items of each Hamiltonian path were ignored and analyses were performed on the remaining 12 items in the Hamiltonian paths. Over five runs, 20 participants were presented with 25 sets of these Hamiltonian walks. Each run was Z scored in time and the TRs corresponding to 4s after the stimulus onset were extracted and averaged to represent the brain's response to

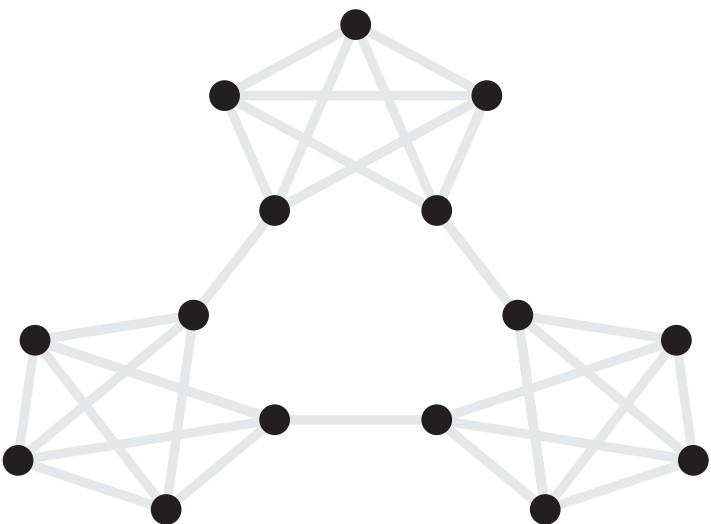

**Figure 4** **Graph of the community structure from** *Schapiro et al. (2013)*. Black dots represent the 15 abstract visual stimuli participants were shown. Grey lines represent possible transitions between the dots.

the stimulus. A "searchlight" approach was used to test which, if any, voxels in the brain represented the community structure. This involves iterating the same computation over a subset ("searchlight") of voxels centered on different voxels in the brain. The searchlight was a spatio-temporal tensor of $3 \times 3 \times 3 \times 15$ voxels centered on every voxel in the brain. In each searchlight the correlation of patterns of voxel activity was computed for all pairs of stimuli, and these were tested to determine whether the correlation for stimuli belonging to the same community was higher than that for stimuli belonging to different communities. A permutation test evaluated the robustness of these metrics across participants. Some regions of the brain, including the inferior frontal gyrus and superior temporal gyrus, reliably showed a greater correlation between stimuli within a community compared to stimuli in different communities.

### Methods

Five steps were performed to simulate the data from *Schapiro et al. (2013)*: generate a template of average voxel activity; extract the noise parameters using fmrisim; represent the timing of each stimulus onset; implement a region of interest mask of the significant voxels; and characterize the signal. All of these were performed using the aforementioned dataset, and then the nature and magnitude of the signal was manipulated as described below.

We embedded this signal in a region of interest (ROI), identified by *Schapiro et al. (2013)* spanning the left superior temporal gyrus, as well as the anterior temporal lobe and inferior frontal gyrus (totaling 442 voxels). The simulated signal was generated with an event-related design in which the different stimuli on the community structure graph (Fig. 4) occurred one after the other. Each stimulus evoked a positive pattern of activity across all the voxels in the ROI. This pattern was determined by that stimulus's position in the

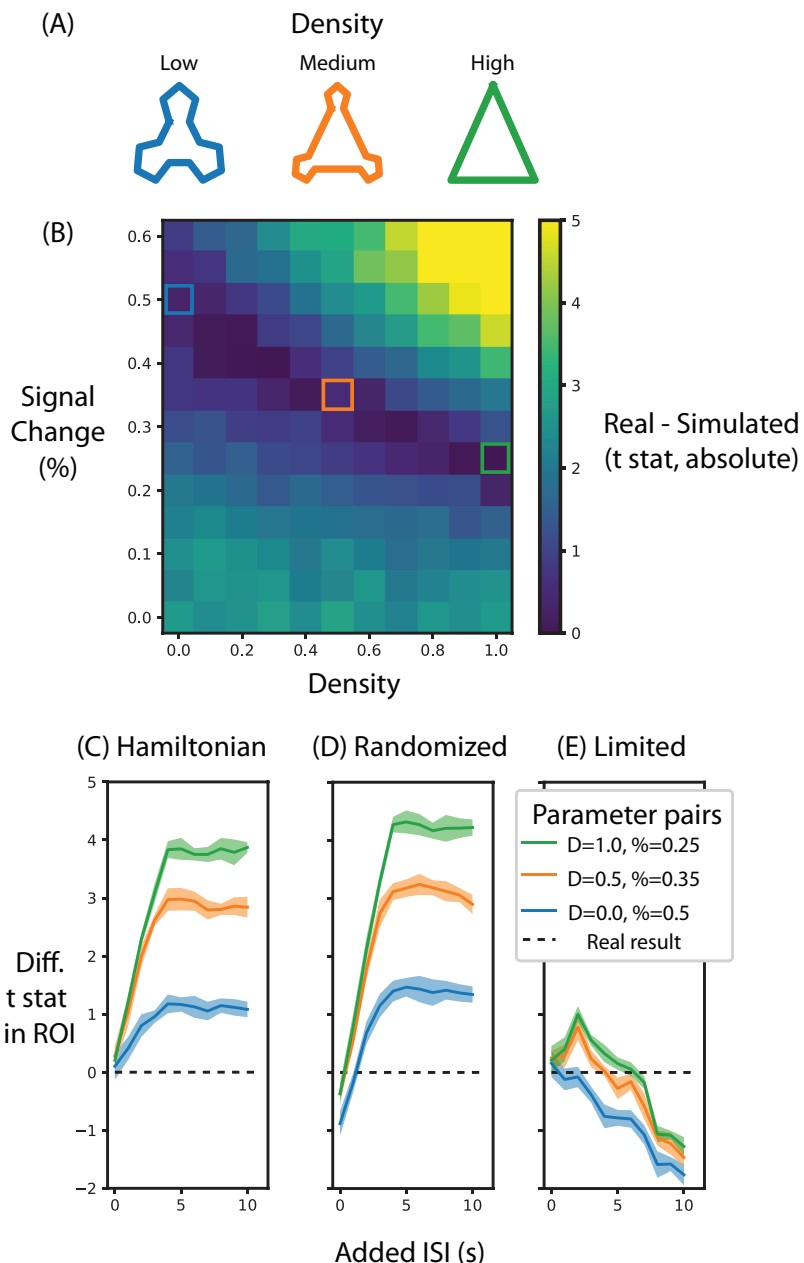

**Figure 5** **Alternative designs for *Schapiro et al. (2013)* and their relative test statistic.** (A) Depictions of representations corresponding to different densities. (B) A heat map of the difference between the real data and simulated data for different parameters of signal magnitude and density. Three pairs of parameters were chosen that minimize the difference between the simulated and real data. (C) For these three pairs of parameters, the average *t*-statistic in the ROI that was significant in *Schapiro et al. (2013)* was subtracted from the real data and compared at different ISIs when the stimuli were presented in a Hamiltonian sequence. The black line represents the real data. (D) is the same as (C) except that the sequence of stimuli was randomized in the simulation. (E) is the same as (C) except that the maximum duration of the run is limited, regardless of ISI. Shaded lines represent the standard deviation across the 10 permutations of this condition. 'D' in the legend refers to the density of the representation, '%' refers to the percent signal change.

high-dimensional embedding of the community structure. In particular, we generated a 2-dimensional graph of the community structure, then an orthonormal transformation was used to embed these points in 442-dimensional space (matching the number of ROI voxels) in a way that preserved Euclidean distance. This meant that each of these voxels' activity represented a dimension along which each event could vary and the pattern across the voxels determined the event's representation. The specified response to every event for each voxel was convolved with the double gamma HRF (*Friston et al., 1998*). The time course of the events was determined by the stimulus presentation timing during the Hamiltonian test phases.

There were two critical parameters determining the representation of the community structure: magnitude and density. Magnitude reflects the neural response that each stimulus presentation evokes and this determines the overall range of the coordinate values in the graph. Specifically, percent signal change was used to determine the evoked response. Percent signal change was scaled for each voxel by taking the mean of that voxel's noise time-series, dividing it by one hundred to make it a proportion and then multiplying it by the percent signal change to get the magnitude of the evoked response for that voxel. Since different stimuli evoke different magnitudes of responses, the magnitude of evoked response specified by the above procedure is used to scale the peak stimulus response (i.e., the coordinate furthest from 0) and all other responses for that voxel are scaled in proportion to their response in the embedding.

Density is the amount of clustering within a community. A high-density representation means that all of the stimuli within a community are similar to one another, reflecting strong event segmentation. In the simulation, a density of 1 means that points within a community are completely overlapping, whereas a density of 0 means that all transitions between stimuli result in equal length edges on the graph (Fig. 5A). Importantly, magnitude and density are independent: a participant's representation of the communities can be perfectly separated (high density) but the neural response evoked by each stimulus might be minimal (low magnitude), meaning that the community structure is not measurable due to neural noise. Hence, the combination of both magnitude and density jointly contribute to the degree of community structure that can be measured neurally.

We simulated fMRI data for pairs of magnitude and density parameter values to observe how they compared to the real data. These were processed using the same analysis pipeline, including the metric for calculating within vs. between community distance, used for the real data to get a summary statistic of the degree of community structure in the ROI where a known signal was added. We took the difference in the permuted $t$-test statistic for ROI voxels in the real and simulated participants to get an estimate of simulated signal strength relative to the real data.

In addition to re-creating the design of the experiment, we used fmrisim to manipulate design parameters and examine their impact on effect size. As discussed above, a critical parameter in event-related design is the ISI. We used fmrisim to explore the impact that this may have had on the results. To do this, time was added between events while preserving the trial jittering, ordering, and trial number (e.g., if the original participant experienced a 3s ISI between events then in this simulation more time was inserted to space out these events,

but otherwise nothing else was changed); allowing new data to be simulated based on this time course. A simulation with 0s added to the ISI was a direct simulation of *Schapiro et al. (2013)*. This analysis was run 10 times (i.e., 10 sets of participants were generated for each level of ISI and signal parameter) to estimate the variability of these summary statistics.

We also used fmrisim to examine how the stimulus sequence may have impacted the results; that is, what would have happened if stimuli were not presented in the Hamilitonian path as was done by *Schapiro et al. (2013)*, but rather presented in a random order, irrespective of their community structure (note that a randomized order was undesirable in the experiment because it may have interfered with the learned transition probabilities). For this analysis, the order of stimuli from each Hamiltonian walk was shuffled (such that the same stimuli could never occur consecutively). According to evidence from randomized univariate designs (*Dale, 1999*) this should boost signal greatly.

Finally, although increasing the ISI may boost signal, it also increases the duration of the experiment. To investigate this, we also examined the effect of limiting the number of trials per run. For this analysis, we increased the ISI between events in a Hamiltonian walk, as before, but cut-off the sequence of stimuli when the run ended for the real participants, and analyzed the truncated dataset to determine the influence of this manipulation on power.

### Results

The first aim of these analyses was to determine the range of percent signal change required to replicate the results from *Schapiro et al. (2013)*. To do so, we compared the summary statistic for real and simulated data for different pairs of signal magnitude and density parameters. Figure 5B shows the absolute value of this difference, with low difference scores indicating the simulation more accurately approximated the real test statistic compared to high difference scores. When the simulated community structure is low density, the signal magnitude required to approximate the test statistic of *Schapiro et al. (2013)* is higher than the signal magnitude required when the density is greater. This analysis shows that the necessary magnitude is within the plausible range, regardless of community density. Hence, if this simulation were done before data collection (using default noise parameters from other participants), this analysis would show that this design, in terms of the specified participant number, trial number and ISI, is sufficiently powered to identify community structure. Indeed, if this simulation were run many times it could estimate the likelihood of achieving a significant result, akin to the use of a traditional power analysis for calculating the probability of success.

We chose three pairs of magnitude and density parameters that lie along the trough of Fig. 5B (i.e., parameters that minimize the difference in test statistic between the real and simulated data) and varied the ISI to observe the effect it has on the test statistic, as shown in Fig. 5C. The peak test statistic is greatest with a minimum ISI of 5s. Critically, the test statistic plateaus after this ISI, suggesting that a slower event-related design would confer no additional benefit, especially given the cost of additional experimental time. This pattern is present regardless of the signal density, although a denser signal structure has a higher test statistic.

Surprisingly, and counter to dogma conquering univariate analyses, randomizing the order of test stimuli did not greatly change the test statistic. Figure 5D shows that when the minimum ISI is 1s, as in *Schapiro et al. (2013)*, and the stimulus sequences are randomized, then the test statistic is lower than when they are presented in a Hamilitonian order especially with low density representations. In other words, randomizing the stimulus sequence would decrease the test statistic. However, if the ISI for these randomized sequences is increased then the test statistics are only slightly greater than the test statistic for a Hamiltonian sequence. Hence, randomizing would only be beneficial for this experiment design if the ISI were long.

Although increasing ISI increases the signal when the experiment duration is potentially limitless (Figs. 5C and 5D), experiments are practically constrained in terms of how long a session can run. Figure 5E shows that when the experiment duration is constrained to be the same as *Schapiro et al. (2013)*, the test statistic in general decreases. Interestingly, if the density is above zero then there is a slight gain by adding an ISI of 2s but no such benefit exists when the density is 0. This suggests that given the time constraints of fMRI, the design chosen by *Schapiro et al. (2013)* was efficient and nearly optimal. However, such foresight is not always forthcoming as noted in the discussion below.

### Discussion

The aim of the present analyses was to show how fmrisim can help researchers make informed design choices, regarding factors such as ISI and stimulus ordering, to attain the highest statistical power in their design as possible. Across the different signal parameters, the same trends in effect size were found for different ISIs: the effect size increases as the ISI increases up to 5s, after which it plateaus; however, this gain in effect size was greatest when the signal was dense. Counter to typical assumptions in fast event-related designs (*Burock et al., 1998*), a randomized order of stimuli slightly reduced the effect size when the ISI was short and only slightly helped when the ISI was longer. Of course, any increase in ISI needs to be weighed against how this affects experiment duration and thus the number of allowable trials. Others have shown that longer ISIs can help the effect size despite such limits (*Friston et al., 1999*), whereas for this design we observed that increasing ISI resulted in a diminished effect size (*Kriegeskorte, Mur & Bandettini, 2008*), although this was non-monotonic for some signal densities.

The analyses described above provide an example of how fmrisim can validate the plausibility of an experimental analysis; however, fmrisim can also be used to show the implausibility of an experiment/analysis. We have taken this approach in other work using fmrisim to demonstrate what experimental design procedures can and cannot result in signal that can be recovered using a particular analysis procedure—topological data analysis (*Ellis et al., 2019*). Hence we believe that fmrisim can be used to demarcate what experiments and analyses are and are not plausible. The example outlined above also suggests ways in which simulation can be used for pre-registration. For example, consider designing a study similar to that of *Schapiro et al. (2013)*, but that required a change in some parameter(s), such as ISI. The data generation and analysis procedures described above could be used to evaluate the influence of this manipulation on the hypothesized

effect, which can then be pre-registered and made openly accessible both during review and after publication. The hypothesized results and final analyses could also be compared to what was pre-registered and be interpreted accordingly.

In sum, the kinds of simulations supported by fmrisim can be used to improve the power of designs and facilitate open science practices in research.

## CONCLUSION

We have presented a new package, fmrisim, for simulation of realistic fMRI data. To our knowledge, fmrisim is the only package capable of taking in raw fMRI data and generating synthetic fMRI data that is approximately matched in terms of some of the raw data's noise properties. This software is open-source and updates are screened with rigorous tests. Code for all of the analyses and plots included here can be found in online public repositories to allow users to utilize and explore this package for their own research. It is particularly useful that fmrisim is written in Python, a rapidly expanding platform for advanced neuroimaging analysis. We show here that fmrisim can be profitably used to evaluate the effects of design and analysis parameters and facilitate pre-registration of fMRI data. With the accelerating development of advanced neuroimaging methods, fmrisim can help evaluate and compare these methods on a standardized and openly accessible software platform that can facilitate both data analysis and further methods development.

### Funding
This study was supported by Intel Corporation and the John Templeton Foundation. The funders had no role in study design, data collection and analysis, decision to publish, or preparation of the manuscript.

### Grant Disclosures
The following grant information was disclosed by the authors:
Intel Corporation and the John Templeton Foundation.

### Competing Interests
The authors declare there are no competing interests.

### Author Contributions
- Cameron T. Ellis conceived and designed the experiments, performed the experiments, analyzed the data, prepared figures and/or tables, authored or reviewed drafts of the paper, and approved the final draft.
- Christopher Baldassano conceived and designed the experiments, analyzed the data, authored or reviewed drafts of the paper, and approved the final draft.
- Anna C. Schapiro performed the experiments, authored or reviewed drafts of the paper, and approved the final draft.

- Ming Bo Cai analyzed the data, authored or reviewed drafts of the paper, and approved the final draft.
- Jonathan D. Cohen conceived and designed the experiments, authored or reviewed drafts of the paper, and approved the final draft.

## Human Ethics

The following information was supplied relating to ethical approvals (i.e., approving body and any reference numbers):

Princeton University granted ethical approval to carry out the studies that were performed to create this open access data. Data was collected under two protocols approved by the Princeton Institutional Review Board. For the data from Bejjanki et al. (2017), 4486 was the number. For Schapiro et al. (2013), 5048 was the number.

## Data Availability

The following information was supplied regarding data availability: Code for fmrisim is available: https://github.com/brainiak/brainiak/.

Code for a walkthrough of how to use fmrisim is available here: https://github.com/brainiak/brainiak/blob/master/examples/utils/fmrisim_multivariate_example.ipynb.

Code to complete the analyses in the article is available here: https://github.com/CameronTEllis/fmrisim_validation_application.git.

Data for some of the analyses is available here: http://arks.princeton.edu/ark:/88435/dsp01dn39x4181.

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
