# Peer review of "Facilitating open-science with realistic fMRI simulation: validation and application"

_PeerJ, doi:10.7717/peerj.8564_

## Round 0.1 · original submission · Minor Revisions

Your manuscript has now been seen by two reviewers. You will see from their comments below that they find your work of interest, and some constructive points are worth considering. We therefore invite you to revise and resubmit your manuscript, taking into account the points raised.

·

Basic reporting

I couldn't find anything that could be improved in terms of use of English. Regarding the structure, in the description of the first (validation) experiment, P21 L370, different contributions and considerations regarding the simulated signal are repeated (and already mentioned) on P19 and P20.

You might potentially think to restructure the headings to better fit with the journal guidelines - right now there are 3 main sections: fmrisim, validation and application, with their own subsections of intro/methods/results/discussion. These now read as three separate papers.

Experimental design

I really appreciated the ipython notebooks that can be explored.

Validity of the findings

This tool could really help with planning fMRI experiments by offering a way to simulate outcomes depending on sample size, effect size (for some reason always assumed to be 0.8), timing and type of analysis.

As explained in your discussion(s), nonlinear effects are not (yet) included, but this software goes a long way to have a means that assesses effects of noise on fMRI findings. Do you have plans or a timeline to incorporate the nonlinear effects in this tool?

Additional comments

No further comments; I enjoyed reading this

Reviewer 2 ·

Basic reporting

This is a well-written paper which is sufficiently referenced and provides full sharing of all data and code.

Experimental design

This paper introduces fMRI simulation software building on previously published packages with a new focus on multivariate analysis and matching simulated data with real data. I believe the software is discussed in enough detail and validation of the methods is provided as well as a worked through example of how this software will benefit researchers.

Validity of the findings

The validation studies have been carried out with rigour and demonstrate the usefulness and appropriateness of the package. I appreciate the authors being self-critical on the limits of simulation and being very explicit about the assumptions made during the simulation process.

Additional comments

I believe this paper is a useful addition to the literature and complements previous work on simulating fMRI data. My only remark would be around the usability of software for non-expert researchers. Could you please add a section on which skills fMRI researchers require to adopt your software for planning their experiments? I am personally thinking of what level of programming skills, level of detail of understanding of simulation procedures, etc.

---

## Round 0.2 · accepted · Accept

Thank you for the revised manuscript and response letter. I am pleased to inform you that your manuscript has been accepted for publication in PeerJ.